# MDCrow: Automating Molecular Dynamics Workflows with Large Language Models

## Abstract

Molecular dynamics (MD) simulations are essential for understanding biomolecular systems but remain challenging to automate. Recent advances in large language models (LLM) have demonstrated success in automating complex scientific tasks using LLM-based agents. In this paper, we introduce MDCrow, an agentic LLM assistant capable of automating MD workflows. MDCrow uses chain-of-thought reasoning over 40 expert-designed tools for handling and processing files, setting up simulations, analyzing the simulation outputs, and retrieving relevant information from literature and databases. We assess MDCrow's performance across 25 tasks of varying complexity, and we evaluate the agent's robustness to both task complexity and prompt style. `gpt-4o` is able to complete complex tasks with low variance, followed closely by `llama3-405b`, a compelling open-source model. While prompt style does not influence the best models' performance, it may improve performance on smaller models.

## 1 Introduction

Molecular dynamics (MD) simulations have a longstanding role in understanding the behavior of dynamic and complex systems in chemistry and biology. Although MD is an established field, its use in scientific workflows has grown substantially in recent decades (Sinha et al., 2022; Karplus & McCammon, 2002; Hollingsworth & Dror, 2018). This growth is driven by two main factors: (1) MD simulations offer valuable insights into structural and dynamic phenomena, and (2) improved computational hardware and user-friendly software packages have made MD more accessible to a broader range of researchers (Hollingsworth & Dror, 2018). Despite these advances, designing an MD workflow remains challenging. Researchers must select force fields, integrators, simulation lengths, and equilibration protocols, often guided by expert intuition. The process also requires extensive pre- and post-processing, such as preparing protein structures, adding solvents, or analyzing stability under varied conditions.

For a protein simulation, users typically provide a PDB file (Velankar et al., 2021), choose a force field (e.g., CHARMM (Brooks et al., 2009), AMBER (Ponder & Case, 2003)), and set parameters such as temperature, time step, and overall simulation length. They may also clean or trim the structure, add ions or solvent, and analyze the resulting trajectory—choices that depend on the biochemical system and the research goals. Although various tools automate parts of MD workflows or target specific niches (Baumgartner & Zhang, 2020; Hayashi et al., 2022; Singh et al., 2023; Ribeiro et al., 2018; Gygli & Pleiss, 2020; Yekeen et al., 2023; Maia et al., 2020; Ganguly et al., 2022; Rêgo et al., 2022; Groen et al., 2016; Carvalho Martins et al., 2021; Suruzhon et al., 2020), a truly domain-agnostic solution remains elusive. Community-driven toolkits (e.g., EasyAmber (Suplatov et al., 2020), PACKMOL (Martínez et al., 2009), MDAnalysis (Michaud-Agrawal et al., 2011), MDTraj (McGibbon et al., 2015), OpenMM (Eastman et al., 2017), GROMACS (Abraham et al., 2015), LAMMPS (Thompson et al., 2022), SimStack (Rêgo et al., 2022)) and visualization interfaces (Goret et al., 2017; Ribeiro et al., 2018; Rusu et al., 2014; Hildebrand et al., 2019; Biarnés et al., 2012; Humphrey et al., 1996; Sellis et al., 2009; Martínez-Rosell et al., 2017; Ribeiro et al.,

---

†These authors contributed equally to this work
*Corresponding author: andrew.white@rochester.edu

2018) have improved accessibility, but the high variability of MD workflows continues to impede full automation.

Large-Language Model (LLM)-powered agents (Schick et al., 2023; Karpas et al., 2022; Yao et al., 2022; Narayanan et al., 2024) offer a new approach for automating technical tasks by leveraging reasoned tool usage, and have shown promise in chemical synthesis (Bran et al., 2023; Boiko et al., 2023; Villarreal-Haro et al., 2023), materials research (Jablonka et al., 2023; Su et al., 2024; Chiang et al., 2024; Kim et al., 2024), and data aggregation (Lee et al., 2024; Skarlinski et al., 2024).

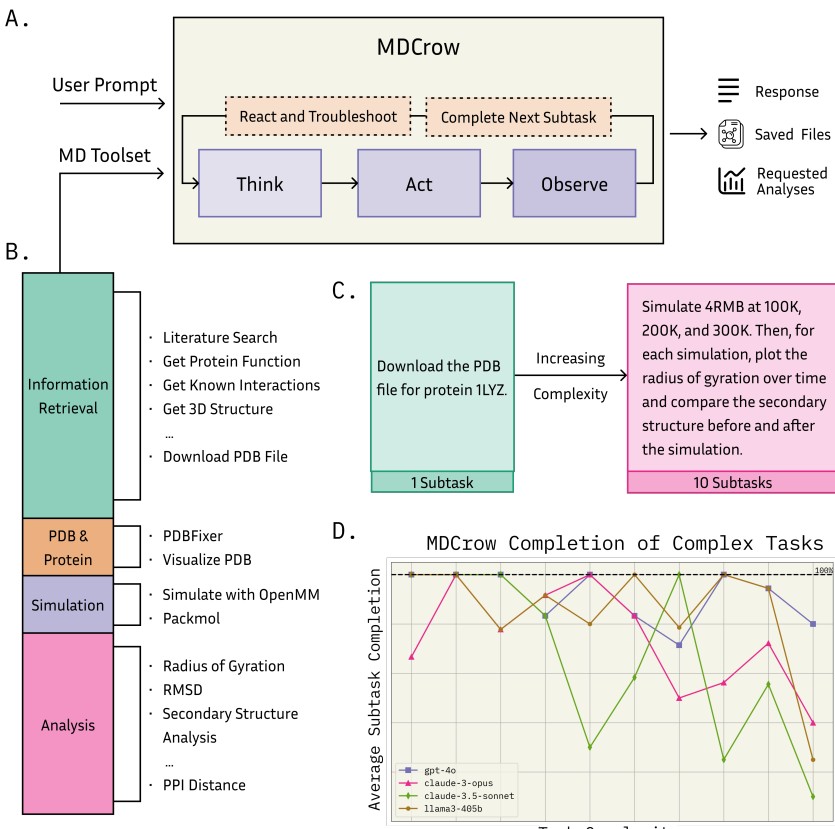

Figure 1: **A.** MDCrow's chain-of-thought workflow starts with a user prompt, uses a set of MD tools, and completes each subtask before producing a final response, along with relevant analyses and files. **B.** Tool usage falls into four categories: information retrieval, PDB/protein handling, simulation, and analysis. Representative tools from each category are shown. **C.** Two example prompts tested with MDCrow: one with a single subtask and one with 10 subtasks. **D.** Average subtask completion across all 25 prompts versus task complexity. Among the top three base-LLMs, `gpt-4o` and `llama3-405b` maintain high completion rates, staying near 100% even as complexity rises.

Here, we introduce MDCrow, an LLM-agent capable of autonomously completing MD workflows in biochemical contexts. We evaluate MDCrow across 25 tasks of varying difficulty and compare its performance using different base models (e.g., `gpt-4o` or `llama3-405b`). We also measure robustness to prompt style and task complexity, and benchmark MDCrow against both single-query LLM approaches and a ReAct-style LLM-agent equipped with a Python interpreter. In all cases, MDCrow outperforms these alternatives (see Figure **1D**). By bringing together reasoning, tool usage, and adaptability, MDCrow addresses a longstanding need for a fully autonomous MD agent—one that can lower the barrier to entry for novices while streamlining the workflow for experts.

## 2 METHODS

### 2.1 MDCROW TOOLSET

MDCrow is an LLM-powered agent built on Langchain (Chase, 2022), which follows a chain-of-thought reasoning process to complete complex tasks (Figure **1A**). We focus the simulation and analysis toolsets in this study on the OpenMM (Eastman et al., 2017) and MDTraj (McGibbon et al., 2015) packages, but it is important to note that this framework is generalizable to any package, provided the appropriate tools. MDCrow's tools can be categorized in four groups: Information Retrieval, PDB & Protein, Simulation, and Analysis (see Figure **1B**).

**Information Retrieval Tools**   These tools handle context-building and quick user queries, including wrappers for UniProt API (UniProt Consortium, 2022) to access protein data and a Literature-Search tool based on PaperQA (Skarlinski et al., 2024) for relevant PDFs (details in the Supplementary Information). Such data can guide parameter selection or simulation strategies.

**PDB & Protein Tools**   MDCrow uses these tools to interact directly with PDB files, performing tasks such as cleaning structures with PDBFixer (Eastman et al., 2017), retrieving PDBs for small molecules and proteins, and visualizing PDBs through Molrender (Developers, 2019) or NGLview (Nguyen et al., 2018).

**Simulation Tools**   OpenMM (Eastman et al., 2017) is used for simulation, while PackMol (Martínez et al., 2009) handles solvent addition. The tools detect incomplete pre-processing or missing parameters, and MDCrow can revise simulation scripts if errors arise. These tools ultimately generate Python scripts that MDCrow can edit on the fly.

**Analysis Tools**   This group of tools is the largest in the toolset, designed to cover common MD workflow analysis methods, with many built on MDTraj (McGibbon et al., 2015) functionalities. Examples include computing the root mean squared distance (RMSD) with respect to a reference structure, analyzing the secondary structure, and various plotting functions.

## 3 RESULTS

### 3.1 MDCROW PERFORMANCE ON VARIOUS TASKS

We evaluated MDCrow on 25 tasks, each requiring between 1 and 10 subtasks. For example, the simplest prompt needed just one step, while a complex prompt involved downloading a PDB file, running three simulations, and performing multiple analyses. MDCrow could perform extra actions without penalty but was penalized for omitting required subtasks. These 25 prompts were tested across three GPT models (`gpt-3.5-turbo-0125`, `gpt-4-turbo-2024-04-09`, `gpt-4o-2024-08-06`), two Llama models (`llama-v3p1-405b-instruct`, `llama-v3p1-70b-instruct`), and two Claude models (`claude-3-opus-20240229`, `claude-3-5-sonnet-20240620`). A newer Claude model (`claude-3-5-sonnet-20241022`) showed no improvement and was not included in these tests.

All parameters except the model choice remained the same, and each MDCrow version ran each prompt only once. Expert evaluators recorded how many subtasks were completed correctly, noting whether a run contained inaccuracies, runtime errors, or hallucinations. Accuracy was judged based on consistency with the expected workflow rather than a fixed reference solution, acknowledging that agent trajectories may vary even when tasks are successfully completed.

We also compared MDCrow against two baselines: (1) a ReAct (Yao et al., 2022) agent with a Python REPL tool and (2) a single-query LLM. All were tested on the same 25 prompts with `gpt-4o`. We provided different system prompts to align each framework with MDCrow's tool stack (details in Supplemental Information). The single-query LLM generated code for all subtasks, while the ReAct agent wrote and executed code using a chain-of-thought approach.

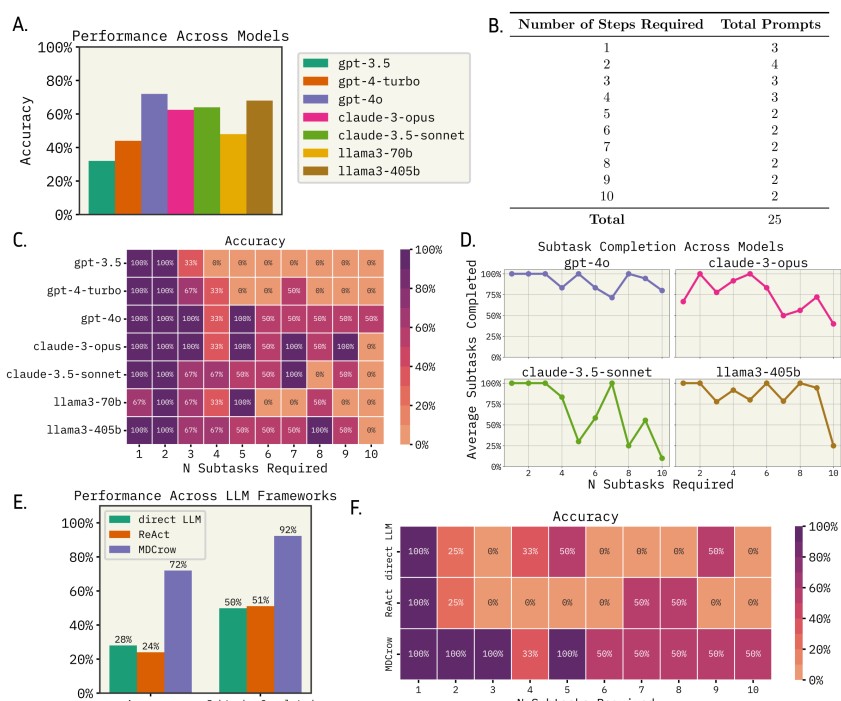

Figure 2: MDCrow performance across different Large Language Models. **A.** Accuracy (acceptable final answer) by LLM. `gpt-4o` outperforms other GPT models significantly ($0.004 \leq$ p-value $\leq 0.046$) but does not differ significantly from Claude or Llama models. **B.** Distribution of subtasks (1–10) across 25 prompts. Each step count is used in at least two prompts. **C.** Percentage of accurate solutions vs. subtask count for each LLM. All models show a significant negative correlation between accuracy and task complexity ($3.9 \times 10^{-7} \leq$ p-value $\leq 1.1 \times 10^{-2}$). **D.** Percentage of subtasks completed by MDCrow for top four LLMs across all tasks. **E.** Performance among LLM frameworks, using `gpt-4o`. MDCrow is more accurate and completes more subtasks than direct LLM and ReAct with only Python REPL tool. **F.** Percentage of accurate solutions vs. subtask count for each LLM framework type. All show a significant negative correlation between accuracy and task complexity ($1 \times 10^{-4} \leq$ p-value $\leq 7 \times 10^{-2}$)

MDCrow outperformed both baselines by a notable margin in completing subtasks and producing accurate solutions (Figure **2E**). While the baseline performance quickly dropped to near-zero after just three steps, MDCrow sustained more reliable performance across the full complexity range, aided by robust file handling, simulation setup, and the capacity to recover from errors.

## 3.2 MDCROW ROBUSTNESS

We tested MDCrow's robustness on increasingly complex prompts and different prompt styles. To explore how well each model handled growing complexity, we created 10 prompts that successively added subtasks. Each prompt was tested twice: once in a conversational style and once with explicit step-by-step instructions. We then calculated the coefficient of variation (CV) for the percentage of completed subtasks across all tasks. A lower CV means more consistent performance and thus higher robustness. Results showed marked differences among models and prompt types: `gpt-4o` and `llama3-405b` demonstrated moderate to high robustness, while the Claude models scored notably lower (see Figure **3C**).

## 4 DISCUSSION

Although LLMs' scientific abilities are growing (Jaech et al., 2024; Hurst et al., 2024; Laurent et al., 2024), they cannot yet independently complete MD workflows, even with a ReAct framework and Python interpreter. However, with frontier LLMs, chain-of-thought reasoning, and an expert-curated

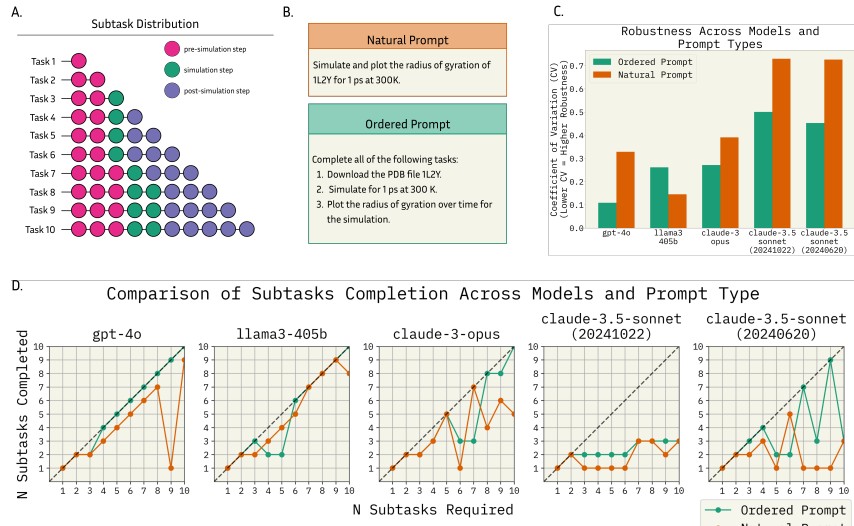

Figure 3: **A**. Tasks categorized by subtask count, starting with one subtask (*Download a PDB file*) and increasing to 10. **B**. Examples of "Natural" vs. "Ordered" prompts for a three-step task. **C**. Robustness (coefficient of variation, CV) of each model and prompt style: lower CV indicates more consistent performance. `gpt-4o` and `llama3-405b` are more robust, while Claude models have higher CVs. **D**. Subtask completion comparison across models and prompt types. In the 9-subtask prompt, `gpt-4o` exited after an early error. Overall, `gpt-4o` and `llama3-405b` handle complexity better, while `claude-3-opus` and `claude-3.5-sonnet` struggle, especially with complex tasks.

toolset, MDCrow successfully handles a broad range of tasks. It performs almost 180% better than `gpt-4o` in ReAct workflows, which is expected due to MD workflows' need for file handling, error management, and real-time data retrieval.

For all LLMs, task accuracy and subtask completion drop as task complexity increases. `gpt-4o` can handle multiple steps with relatively low variance, followed closely by `llama3-405b`, an open-source model. Other models, such as `gpt-3.5` and `claude-3.5-sonnet`, struggle with hallucinations or inability to follow complex instructions. Performance on these models, however, is improved with explicit prompting.

These tasks were focused on routine applications of MD with short simulation runtimes, limited to proteins, common solvents, and force fields included in the OpenMM package. We did not explore small-molecule force fields, especially related to ligand binding. Future work could explore multi-modal approaches (Wang et al., 2024; Gao et al., 2023) for tasks like convergence analysis or plot interpretations. The current framework relies on human-created tools, but as LLM-agent systems become more autonomous (Wang et al., 2023), careful evaluation and benchmarking will be essential.

## 5 CONCLUSION

MDCrow uses LLMs' automation and reasoning capabilities through conversational agents for diverse MD tasks. MDCrow, built on `gpt-4o` or `llama3-405b`, consistently exhibits robust performance across task complexities and prompt variations. While MD automation remains a significant challenge, MDCrow offers an adaptable and user-friendly solution, underscoring the potential for LLM-based agents to further improve MD automation with minimal errors.

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

# A  APPENDIX

## A.1  MEMORY

A key challenge in developing an automated MD assistant is ensuring it can manage a large number of files, analyses, and long simulations and runtimes. Although MDCrow has been primarily tested with shorter simulations, it is designed to handle larger workflows as well. Its ability to retrieve and resume previous runs allows users to start a simulation, step away during the long process, and later continue interactions and analyses without needing to stay engaged the entire time. An example of this memory feature is shown in Figure **4**.

Memory is an optional feature that creates an LLM-generated summary of the user prompt and agent trace, which is assigned to a unique run identifier provided at the end of the run (but accessible at any time during the session). Each run's files, figures, and path registry are saved in a unique checkpoint folder linked to the run identifier.

When resuming a chat, the LLM loads the summarized context of previous steps and maintains access to the same file corpus, as long as the created files remain intact. To resume a run, the user simply provides the checkpoint directory and run identifier. MDCrow then loads the corresponding memory summaries and retrieves all associated files, enabling seamless continuation of analyses.

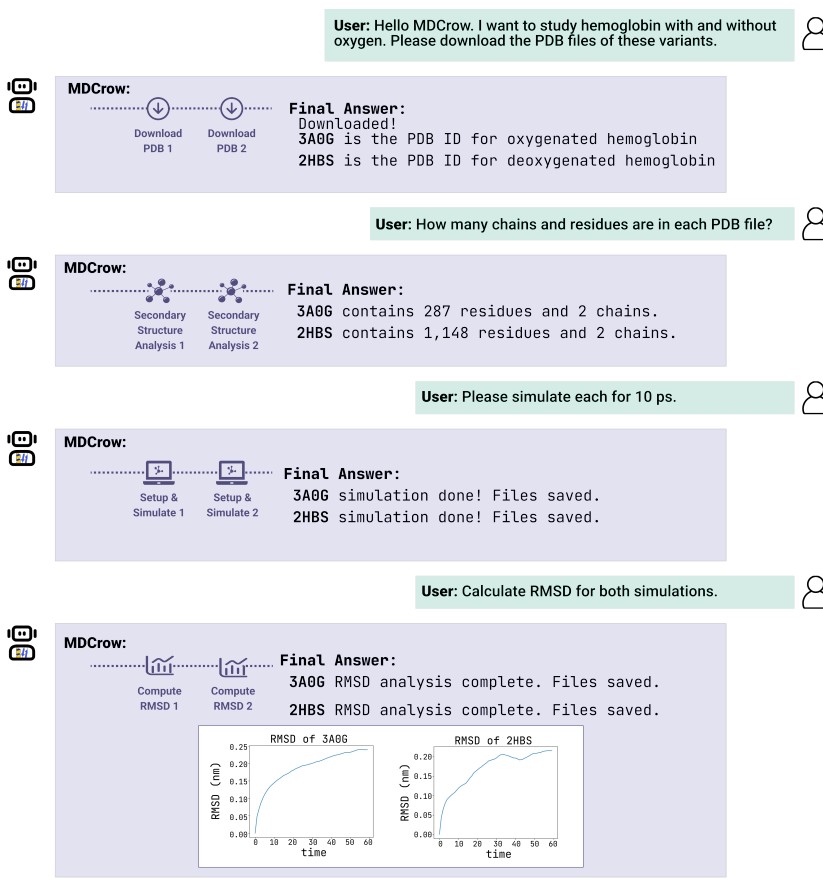

Figure 4: **Example Chat** Example of chat with MDCrow. The user first asks to download PDB files for two systems. Then, once MDCrow has completed this task, the user asks for analysis of the files. Next, the user asks for a quick 10 ps simulation of both files, and MDCrow saves all files for later handling. Lastly, the user asks for plots of RMSD for each simulation over time, and MDCrow responds with each plot.

## A.2 CLAUDE-SPECIFIC ENGINEERING

While both of Claude's Sonnet models achieved poor performance during the robustness experiment, it can be noted that a single common error arose consistently. When running an NPT simulation, MDCrow requires that all parameters be passed to the simulation tool. However, both Sonnet models consistently neglected to provide a value for pressure, even when directly prompted to do so. The `claude-3-opus` made this mistake a single time. This is a relatively simple fix, providing MDCrow with a default pressure of 1 atm when no pressure is passed.

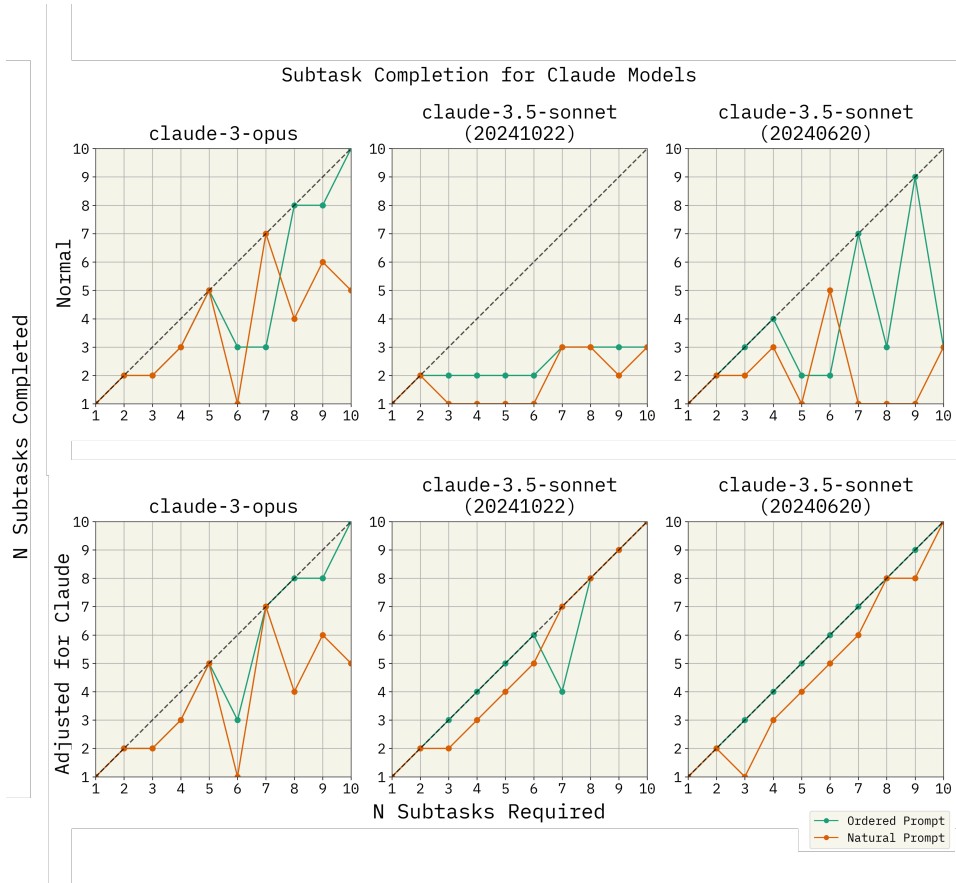

Figure 5: Performance of MDCrow with three Claude models on 10 tasks. As the number of subtasks increase, we all subtasks completed for both prompt types. The top row shows MD-Crow's performance as-is, and the bottom row shows MDCrow's performance when given a direct fix for missing parameters. There is a clear change in performance after the fix for both `claude-3.5-sonnet-20241022` and `claude-3.5-sonnet-20240620`.

As can be seen in Figure 5, including this fix drastically improves the performance of these models, with performance comparable to the top models. However, no other models made this mistake, and no other model-specific optimization was conducted. Thus, for all experiments shown in this paper, MDCrow does not accommodate this Claude-specific missing parameter fix.

## A.3 MDCROW EXTRAPOLATION

We further show MDCrow's ability to harness its memory feature and extrapolate outside of its toolset to complete new tasks. This task requires MDCrow to perform an annealing simulation, which is not part of the current toolset. The agent achieves this by first setting up a simulation to find appropriate system parameters and handle possible early errors. Then, the agent modifies the script according to the user's request. Once the simulation is complete, the user later asks for simulation analyses, shown in Figures **6A,B**.

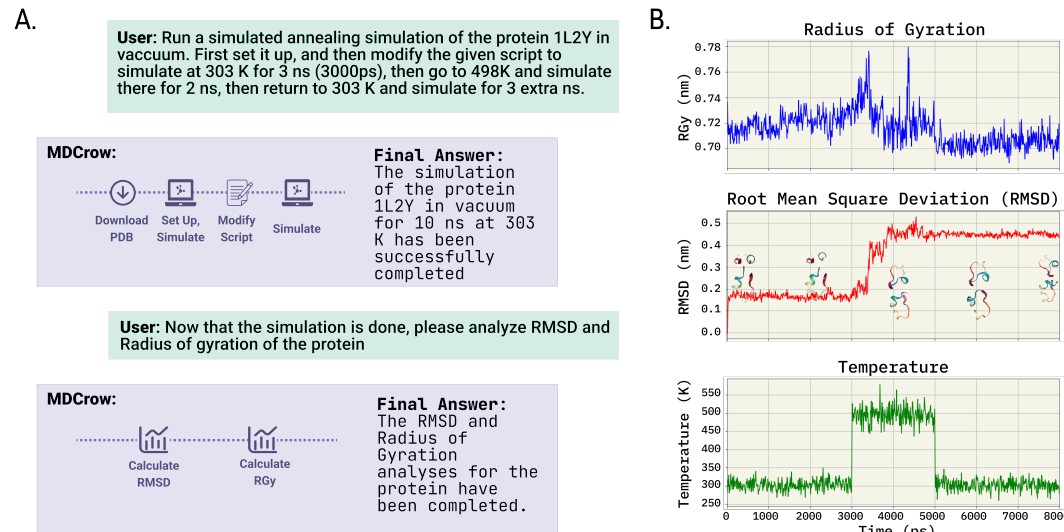

Figure 6: **A.** MDCrow simulating annealing. The user directly instructs to simulate an annealing simulation of protein 1L2Y. The user then utilizes the memory feature to ask for further analyses. **B.** RMSD, RGy, and temperature throughout the simulation, as requested by the user in A.

### A.4    PROMPTS

**MDCrow Prompt**

```
You are an expert molecular dynamics scientist, and your task is to
respond to the question or solve the problem to the best of your ability
using the provided tools.

You can only respond with a single complete 'Thought, Action, Action
Input' format OR a single 'Final Answer' format.

Complete format:
Thought: (reflect on your progress and decide what to do next)
Action:
```
{{
    "action": (the action name, it should be the name of a tool),
    "action_input": (the input string for the action)
}}
```

OR

Final Answer: (the final response to the original input
question, once all steps are complete)

You are required to use the tools provided, using the most specific tool
available for each action. Your final answer should contain all
information necessary to answer the question and its subquestions.
Before you finish, reflect on your progress and make sure you have
addressed the question in its entirety.

If you are asked to continue or reference previous runs, the context
will be provided to you. If context is provided, you should assume you
are continuing a chat.

Here is the input:
Previous Context: {context}
Question: {input}
```

During the comparison study between MDCrow, GPT-only, and ReAct with Python REPL tool, we used different system prompts for each of these LLM frameworks.

**Direct-LLM Prompt**

```
You are an expert molecular dynamics scientist, and your task is to
respond to the question or solve the problem in its entirety to the best
of your ability. If any part of the task requires you to perform an
action that you are not capable of completing, please write a runnable
Python script for that step and move on. For literature papers, use and
process papers from the 'paper_collection' folder. For .pdb files,
download them from the RSCB website using 'requests'. To preprocess PDB
files, you will use PDBFixer. To get information about proteins,
retrieve data from the UniProt database. For anything related to
simulations, you will use OpenMM, and for anything related to analyses,
you will use MDTraj. At the end, combine any scripts into one script.
```

**ReAct Agent Prompt**

```
You are an expert molecular dynamics scientist, and your task is to
respond to the question or solve the problem to the best of your
ability. If any part of the task requires you to perform an action that
you are not capable of completing, please write a runnable Python script
for that step and run it. For literature papers, use and process papers
from the 'paper_collection' folder. For .pdb files, download them from
the RSCB website using 'requests'. TO preprocess PDB files, you will use
PDBFixer. To get information about proteins, retrieve data from the
UniProt database. For anything related to simulations, you will use
OpenMM, and for anything related to analyzes, you will use MDTraj.

You can only respond with a single complete 'Thought, Action, Action
Input' format OR a single 'Final Answer' format.

Complete format:
Thought: (reflect on your progress and decide what to do next)
Action:
```
{{
    "action": (the action name, it should be the name of a tool),
    "action_input": (the input string for the action)
}}
```

OR

Final Answer: (the final response to the original input
question, once all steps are complete)

You are required to use the tools provided,
using the most specific tool available for each action. Your final
answer should contain all information necessary to answer the question
and its subquestions. Before you finish, reflect on your progress and
make sure you have addressed the question in its entirety.

Here is the input:
Question: {input}
```

## A.5 EXPERIMENT TASK PROMPTS & REFERENCES USED IN EXPERIMENTS

Table 1: Details of 25 task prompts used in experiments

| Prompt ID | Prompt | # sub-tasks | List of required sub-tasks |
|---|---|---|---|
| 1 | Simulate PDB ID 1MBN at two different temperatures: 300 K and 400 K for 1 ns each. Plot the RMSD of both over time and compare the final secondary structures at the end of the simulations. Get information about this protein, such as the number of residues and chains, etc. | 8 | Download PDB, simulate (x2), RMSD (x2), DSSP (x2), summarize_protein |
| 2 | Download the PDB file for protein 1LYZ. | 1 | Download PDB |
| 3 | Download the PDB file for protein 1GZX. Then, analyze the secondary structure of the protein and provide information on how many helices, sheets, and other components are present. Get the gene names for this protein. | 3 | Download PDB, DSSP, GetProteinFunction (or literature) |
| 4 | What are the common parameters used to simulate fibronectin? | 1 | literature search |
| 5 | Simulate 1VII for 1 ns at a temperature of 300 K. Then, tell me if the secondary structure changed from the beginning of the simulation to the end of the simulation. | 5 | Download PDB, simulate, DSSP before, DSSP after, comparison |
| 6 | Simulate 1A3N and 7VDE (two PDB IDs matching hemoglobin) with identical parameters. Find the appropriate parameters for simulating hemoglobin from the literature. Then, plot the radius of gyration throughout both simulations. | 7 | Download PDB (x2), literature, simulate (x2), RGy (x2) |
| 7 | Simulate 1ZNI for 1 ns at a temperature of 300 K in water. Then, simulate it again in acetonitrile. Compute the RMSD, final secondary structure, and PCA for each simulation. | 10 | Download PDB, simulate (x2), packmol, RMSD (x2), SASA (x2), PCA (x2) |
| 8 | Simulate 4RMB at 100K, 200K, and 300K. Then, for each simulation, plot the radius of gyration over time and compare the secondary structure before and after the simulation. | 10 | Download PDB, simulate (x3), RGy (x3), DSSP (x3) |
| 9 | Download the PDB file for 1AEE. Then tell me how many chains and atoms are present in the protein. | 2 | download PDB, count atoms/chains |
| 10 | Simulate protein 1ZNI at 300 K for 1 ns and calculate the RMSD. | 3 | Download PDB, simulate, RMSD |
| 11 | Download the PDB files for 8PFK and 8PFQ. Then, compare the secondary structures of the two proteins, including the number of atoms, secondary structures, number of chains, etc. | 4 | Download PDB (x2), DSSP (x2) |
| 12 | Simulate fibronectin (PDB ID 1FNF) for 1 ns, using an appropriate temperature found in the literature. Compute the RMSD and the final secondary structure. By using the PDB ID to get the Uniprot ID, obtain the subunit structure and the number of beta sheets, helices, etc. Compare this information to the structure we computed. | 8 | Download PDB, literature, simulate, RMSD, DSSP, get uniprot, subunit structure, get beta sheets/helices |
| 13 | Compare the RMSF of 1UBQ under high pressure and low pressure. Perform the simulation for 1 ns, varying only the pressure. Plot the moments of inertia over time for both simulations. | 7 | Download PDB, simulate (x2), RMSF (x2), MOI (x2) |
| 14 | Simulate deoxygenated hemoglobin (1A3N) and oxygenated hemoglobin (6BB5). Plot the PCA of both trajectories. | 6 | Download PDB (x2), simulate (x2), PCA (x2) |

| Prompt ID | Prompt | # sub-tasks | List of required sub-tasks |
|---|---|---|---|
| 15 | Simulate trypsin (1TRN) for 1 ns at 300 K and plot eneRGy over time. Compute SASA, RMSF, and radius of gyration. Get the subunit structure, sequence, active and binding sites. | 9 | Download PDB, simulate, output figures, SASA, RMSF, RGy, subunit structure, sequence info, all known sites |
| 16 | Download the PDB file for 1C3W and describe the secondary structure. Then, simulate the protein at 300 K for 1 ns. Plot the RMSD over time and the radius of gyration over time. | 5 | Download PDB, DSSP, simulate, RMSD, RGy |
| 17 | Download the PDB file for 1XQ8, and then save the visualization for it. | 2 | Download PDB, visualize |
| 18 | Download the PDB for 2YXF. Tell me about its stability as found in the literature. Then, simulate it for 1 ns and plot its RMSD over time. | 4 | Download PDB, literature search, simulate, RMSD |
| 19 | Simulate 1MBN in water and methanol solutions. | 4 | Download PDB, packmol to get appropriate non-water solvent, simulate (x2) |
| 20 | Download protein 1ATN. | 1 | Download PDB |
| 21 | Download and clean protein 1A3N. | 2 | Download PDB, clean |
| 22 | Perform a brief simulation of protein 1PQ2. | 2 | Download PDB, simulate |
| 23 | Analyze the RDF of the simulation of 1A3N solvated in water. | 3 | Download PDB, simulate, RDF |
| 24 | Simulate oxygenated hemoglobin (1A3N) and deoxygenated hemoglobin (6BB5). Then analyze the RDF of both. | 6 | Download PDB (x2), simulate (x2), RDF (x2) |
| 25 | Simulate 1L6X at pH 5.0 and 8.8, then analyze the SASA and RMSF under both pH conditions. | 9 | Download PDB, clean at pH 5.5 and 8.0, simulate(x2), SASA(x2), RMSF(x2) |

**List of References Used for Literature Search During the Experiments.**

1. The folding space of protein $\beta$2-microglobulin is modulated by a single disulfide bridge, `10.1088/1478-3975/ac08ec`

2. Molecular Dynamics Simulation of the Adsorption of a Fibronectin Module on a Graphite Surface, `10.1021/la0357716`

3. Predicting stable binding modes from simulated dimers of the D76N mutant of $\beta$2-microglobulin, `10.1016/j.csbj.2021.09.003`

4. Deciphering the Inhibition Mechanism of under Trial Hsp90 Inhibitors and Their Analogues: A Comparative Molecular Dynamics Simulation, `10.1021/acs.jcim.9b01134`

5. Molecular modeling, simulation and docking of Rv1250 protein from Mycobacterium tuberculosis, `10.3389/fbinf.2023.1125479`

6. Molecular Dynamics Simulation of Rap1 Myb-type domain in Saccharomyces cerevisiae, `10.6026/97320630008881`

7. A Giant Extracellular Matrix Binding Protein of Staphylococcus epidermidis Binds Surface-Immobilized Fibronectin via a Novel Mechanism, `10.1128/mbio.01612-20`

8. High Affinity vs. Native Fibronectin in the Modulation of $\alpha$v$\beta$3 Integrin Conformational Dynamics: Insights from Computational Analyses and Implications for Molecular Design, `10.1371/journal.pcbi.1005334`

9. Forced unfolding of fibronectin type 3 modules: an analysis by biased molecular dynamics simulations, `10.1006/jmbi.1999.2670`

10. Adsorption of Fibronectin Fragment on Surfaces Using Fully Atomistic Molecular Dynamics Simulations, `10.3390/ijms19113321`

11. Fibronectin Unfolding Revisited: Modeling Cell Traction-Mediated Unfolding of the Tenth Type-III Repeat, `10.1371/journal.pone.0002373`

12. Tertiary and quaternary structural basis of oxygen affinity in human hemoglobin as revealed by multiscale simulations, `10.1038/s41598-017-11259-0`

13. Oxygen Delivery from Red Cells, `10.1016/s0006-3495(85)83890-x`

14. Molecular Dynamics Simulations of Hemoglobin A in Different States and Bound to DPG: Effector-Linked Perturbation of Tertiary Conformations and HbA Concerted Dynamics, `10.1529/biophysj.107.114942`

15. Theoretical Simulation of Red Cell Sickling Upon Deoxygenation Based on the Physical Chemistry of Sickle Hemoglobin Fiber Formation, `10.1021/acs.jpcb.8b07638`

16. Adsorption of Heparin-Binding Fragments of Fibronectin onto Hydrophobic Surfaces, `10.3390/biophysica3030027`

17. Mechanistic insights into the adsorption and bioactivity of fibronectin on surfaces with varying chemistries by a combination of experimental strategies and molecular simulations, `10.1016/j.bioactmat.2021.02.021`

18. Anti-Inflammatory, Radical Scavenging Mechanism of New 4-Aryl-[1,3]-thiazol-2-yl-2-quinoline Carbohydrazides and Quinolinyl[1,3]-thiazolo[3,2-b][1,2,4]triazoles, `10.1002/slct.201801398`

19. Trypsin-Ligand binding affinities calculated using an effective interaction entropy method under polarized force field, `10.1038/s41598-017-17868-z`

20. Ubiquitin: Molecular modeling and simulations, `10.1016/j.jmgm.2013.09.006`

21. Valid molecular dynamics simulations of human hemoglobin require a surprisingly large box size, `10.7554/eLife.35560`

22. Multiple Cryptic Binding Sites are Necessary for Robust Fibronectin Assembly: An In Silico Study, `10.1038/s41598-017-18328-4`

23. Computer simulations of fibronectin adsorption on hydroxyapatite surfaces, `10.1039/c3ra47381c`

24. An Atomistic View on Human Hemoglobin Carbon Monoxide Migration Processes, `10.1016/j.bpj.2012.01.011`

25. Best Practices for Foundations in Molecular Simulations [v1.0], `10.33011/livecoms.1.1.5957`

26. Unfolding Dynamics of Ubiquitin from Constant Force MD Simulation: Entropy-Enthalpy Interplay Shapes the Free-Energy Landscape, `10.1021/acs.jpcb.8b09318`

27. Dissecting Structural Aspects of Protein Stability

28. MACE Release 0.1.0 Documentation

