# OpenReview forum: "MDCROW: AUTOMATING MOLECULAR DYNAMICS WORKFLOWS WITH LARGE LANGUAGE MODELS"
_ICLR.cc/2025/Workshop/AgenticAI — ICLR 2025 Workshop AgenticAI Poster_

### Official Review · Reviewer_ew4q · 2025-03-03

**Rating:** 7
**Confidence:** 4

**Review:**

This work presents a agentic framework for automating molecular dynamics workflows.

Strength:
- This paper is well presented and generally well-written.
- The MDCrow framework's effectiveness is thoroughly validated through comprehensive experiments across a diverse spectrum of task complexities.
- The evaluation methodology is robust, featuring comparative analyses against multiple state-of-the-art large language models, providing meaningful context for the framework's performance gains.

---

### Official Review · Reviewer_prxo · 2025-03-05

**Rating:** 4
**Confidence:** 4

**Review:**

The paper introduces MDCrow, an AI-driven Large Language Model (LLM) agent designed to automate molecular dynamics (MD) workflows. Traditional MD simulations require complex setup, parameter tuning, and extensive manual intervention, making them difficult to streamline. MDCrow leverages chain-of-thought reasoning and integrates with 40 expert-designed tools for protein structure preparation, simulation execution, result analysis, and literature retrieval. The system is evaluated across 25 tasks of varying complexity, demonstrating that GPT-4o and Llama3-405b achieve the highest task completion rates. MDCrow outperforms single-query LLMs and ReAct-style agents, showing superior accuracy and robustness in automating MD workflows.

Weaknesses:
1. The citations in the paper are not proper, and they do not follow the ICLR template.
2. Computationally intensive, requiring high-performance LLMs for optimal performance.
3. Limited adaptability to novel simulations, as they rely on pre-defined toolsets and workflows.

---

### Decision · Program_Chairs · 2025-03-05

Accept (Poster)